# Evaluation of Expressed MicroRNAs as Prospective Biomarkers for Detection of Breast Cancer

**DOI:** 10.3390/diagnostics12040789

**Published:** 2022-03-23

**Authors:** Amal Ahmed Mohamed, Ahmed E. Allam, Ahmed M. Aref, Maha Osama Mahmoud, Noha A. Eldesoky, Naglaa Fawazy, Yasser Sakr, Mohamed Emam Sobeih, Sarah Albogami, Eman Fayad, Fayez Althobaiti, Ibrahim Jafri, Ghadi Alsharif, Marwa El-Sayed, Asmaa Sayed Abdelgeliel, Rania S. Abdel Aziz

**Affiliations:** 1Department of Biochemistry and Molecular Biology, National Hepatology and Tropical Medicine Research Institute, Cairo 11511, Egypt; amalahmedhcp@nhtmri.org; 2Department of Pharmacognosy, Faculty of Pharmacy, Al-Azhar University, Assiut 71524, Egypt; 3Faculty of Biotechnology, Modern Sciences and Arts University (MSA), Cairo 11511, Egypt; a_aref@msa.edu.eg; 4Department of Biochemistry, Faculty of Pharmacy, Egyptian Russian University, Cairo 11511, Egypt; dr_mahamahmoud@eru.edu.eg; 5Department of Biochemistry and Molecular Biology, Faculty of Pharmacy for Girls, Al-Azhar University, Cairo 11511, Egypt; n.yassin_a@azhar.edu.eg; 6Department of Clinical Pathology, National Institute of Diabetes & Endocrinology, Cairo 11511, Egypt; naglafawzy1@gothi.gov.eg (N.F.); yasser.azmy@gothi.gov.eg (Y.S.); 7Medical Oncology Department, National Cancer Institute, Cairo 11511, Egypt; info@gothi.gov.eg; 8Department of Biotechnology, College of Science, Taif University, P.O. Box 11099, Taif 21944, Saudi Arabia; dr.sarah@tu.edu.sa (S.A.); e.esmail@tu.edu.sa (E.F.); faiz@tu.edu.sa (F.A.); i.jafri@tu.edu.sa (I.J.); 9College of Clinical Laboratory Sciences, King Saud bin Abdulaziz University for Health Sciences, Jeddah 21423, Saudi Arabia; sharifg@ksau-hs.edu.sa; 10Department of Microbiology and Immunology, Faculty of Medicine, South Valley University, Qena 83523, Egypt; 11Department of Botany & Microbiology, Faculty of Science, South Valley University, Qena 83523, Egypt; asmaa.elgafari@sci.svu.edu.eg; 12Department of Clinical Pathology, National Cancer Institute, Cairo University, Cairo 11976, Egypt; rania_smansy@nci.cu.edu.eg

**Keywords:** breast cancer patients, MicroRNAs, serum, early detection, sensitivity, specificity

## Abstract

Background: Early detection and screening of breast cancer (BC) might help improve the prognosis of BC patients. This study evaluated the use of serum microRNAs (miRs) as non-invasive biomarkers in BC patients. Methods: Using quantitative real-time polymerase chain reaction, we evaluated the serum expression of four candidate miRs (miR-155, miR-373, miR-10b, and miR-34a) in 99 Egyptian BC patients and 40 healthy subjects (as a control). The miRs expression was correlated with clinicopathological data. In addition, the sensitivity and specificity of the miRs were determined using receiver operating characteristic (ROC) curve analysis. Results: Serum miR-155, miR-373, and miR-10b expression were significantly upregulated (*p* < 0.001), while serum miR-34a was downregulated (*p* < 0.00) in nonmetastatic (M0) BC patients compared to the control group. In addition, serum miR-155 and miR-10b were upregulated in BC patients with large tumor sizes and extensive nodal involvement (*p* < 0.001). ROC curve analysis showed high diagnostic accuracy (area under the curve = 1.0) when the four miRs were combined. Serum miR-373 was significantly upregulated in the human epidermal growth factor 2–negative (*p* < 0.001), estrogen receptor–positive (*p* < 0.005), and progesterone receptor (PR)-positive (*p* < 0.024) in BC patients, and serum miR-155 was significantly upregulated in PR-negative (*p* < 0.001) BC patients while both serum miR-155 and miR-373 were positively correlated with the tumor grade. Conclusions: Circulating serum miR-155, miR-373, miR-10b, and miR-34a are potential biomarkers for early BC detection in Egyptian patients and their combination shows high sensitivity and specificity.

## 1. Introduction

Breast cancer (BC) is one of the most common cancers and the second-most common cause of death by cancer among women. In the United States, the number of newly diagnosed BC cases exceeds 281,000 annually, with ~43,000 deaths [1]. The BC incidence rate varies greatly worldwide, being higher in developed countries compared to developing countries, probably because of lifestyle differences [2]. In Egypt, BC is the most common cancer in women, accounting for 18.9% of the total cancer cases, and is the leading cause of death by cancer, accounting for 29.1% of cancer-related deaths [3].

The Egyptian BC phenotype is notably aggressive compared to the Western phenotype, probably because of the predominance of premenopausal patients and late diagnosis of advanced BC. According to the population-based cancer registry of the Ghrabiah Governorate, Egypt, the median age at diagnosis is 10 years less compared to Western countries [4,5]. Among Arab women, the median age at diagnosis is ~48 years and most of the BC patients are <50 years old [6]. In Egypt, the BC incidence rate is 29.9/100,000 population in the 30–34-year age group [7].

Early BC detection and screening can help improve the prognosis of BC patients. A meta-analysis of several randomized trials showed a significant decrease in the mortality rate of BC by recruiting women for early screening mammography which is considered the current cultural obstacle for young women in Egypt [8,9].

The routine diagnosis of BC among Egyptian women depends on serum tumor markers, such as CA15.3 and carcinoembryonic antigen (CEA). However, these biomarkers have low sensitivity and specificity and, hence, are not recommended by most experts [10]. In contrast, circulating nucleotide biomarkers, such as serum microRNAs (miRs), can be used for the early screening of BC in at-risk individuals. miRs are small, ~19–25-nucleotide non-coding RNA molecules that regulate gene expression at the posttranscriptional level and are involved in tumorigenesis by controlling the cell proliferation, apoptosis, and differentiation that results in malignant transformation. miRs are highly stable in circulation and can resist degradation, heating, and boiling. In addition, they can resist being bound to high-density lipoproteins (HDLs), associating with the Argonaute 2 (Ago2) protein, or packaged into microparticles, such as exosome-like particles and microvesicles [11,12,13]. miRs are always located in fragile chromosomal regions harboring DNA amplifications, deletions, or translocations, and miR expression is often dysregulated during tumorigenesis, including BC, which contributes to tumor progression [14].

Because of the tissue heterogeneity of BC and the limitations of current biomarkers, there is an urgent need for new markers that can deal with a wide variety of clinical samples. Studies have reported the oncogenic role of miRs and their potential use as biomarkers in BC [15,16]. Twenty-nine miRNAs were distinctively identified in BC tissue compared to normal tissue. [17]. It was also shown that some of the miRNAs were upregulated, such as miRNA-155, while others were downregulated, such as miRNA 10b, in BC tissue when compared with normal tissue. Furthermore, some of the identified miRNAs were associated with tumor stage, invasion, and metastasis [18].

In addition, exosomal miRNAs have been reported to be one of the most important agents involved in BC tumorigenesis, metastasis, recurrence, and resistance against chemotherapeutic agents. Exosomal miRNAs are excellent candidates for the diagnosis and treatment of BC [19]. Increased serum level of exosomal miRNA-373 in BC patients compared to healthy volunteers was reported. This increase was associated with receptor-negative tumors, suggesting it could be used as a blood-based biomarker for aggressive BC [20].

Therefore, this study investigated the expression profiles of four serum miRs: miR-155, miR-373, miR-10, and miR-34a. We also evaluated their potential role as non-invasive biomarkers for early BC detection in Egyptian patients.

## 2. Materials and Methods

### 2.1. Patients and Blood Sampling

Primary BC patients and healthy volunteers with no history of the previous diagnosis with malignancy (*n* = 160) were recruited by the National Cancer Institute Cairo University, Egypt, between September 2020 and November 2021 (Figure 1). The inclusion criteria were an age of 18–70 years and histologically proven BC. Exclusion criteria were chemotherapy or radiotherapy administration, as well as metastasis. The final number of participants included was 99 BC patients and 40 healthy women (controls). Samples of venous blood (10 mL) were collected before surgery and chemotherapy and serum was separated from blood by centrifugation and stored at −80 °C for future analysis. The study was approved by the ethical committee of the National Cancer Institute Cairo in accordance with the principles of the Declaration of Helsinki. All included patients provided written informed consent.

### 2.2. RNA Extraction and cDNA Synthesis

miRs were isolated from blood samples using the miRNeasy RNA isolation kit (Qiagen, Germany) according to the manufacturer’s instructions. RNA was eluted in 30 µL of RNase-free water. The RNA quality was determined using NanoDrop2000 (Thermo Fisher Scientific, Waltham, MA, USA). The ratio of absorbance at 260 and 280 nm was used to assess RNA purity, and a ratio of 1.8–2.1 was used as an indicator of purified RNA. Next, 1 µg of miR was used for reverse transcription to complementary DNA (cDNA) using the miScript II RT Kit (Qiagen Corporation, Frederick, MD, USA), according to the manufacturer’s instructions, and immediately stored at −20 °C for polymerase chain reaction (PCR) analysis.

### 2.3. Real-Time PCR Analysis of miRs

Quantitative reverse transcription-polymerase chain reaction (RT-PCR) was performed using the StepOnePlus™ System (Applied Biosystems Inc., Foster, CA, USA). RNA, U6 small nuclear (RNU6), was used as a housekeeping control for normalizing real-time PCR results, as recommended previously [21]. Primers and probes were supplied as part of the TaqMan microRNA Assay Kit (Applied Biosystems) in a total volume of a 20 μL reaction. PCR conditions were as follows: 15 min incubation at 95 °C, 40 cycles for 15 s at 94 °C, annealing at 58 °C for 30 s, and finally, extension at 70 °C for 30 s. The miR expression was measured using the comparative Ct method, as described previously [22]. The fold-change (FC) in miR-373, miR-155, miR-34a, and miR-10b expression was calculated by using the 2^−ΔΔCt^ method for RNU6.

### 2.4. Statistical Analysis

Statistical analysis was performed using the Package for Social Sciences (SPSS) for Windows, version 17 (SPSS Inc., Chicago, IL, USA). Categorical variables were presented as percentages, while continuous variables were presented as mean ± standard deviation (SD). The independent sample *t*-test and Mann–Whitney *U* test were used to examine significant differences in the mean between BC and control groups where appropriate. Receiver operating characteristic (ROC) curve analysis was performed to determine the best cut-off value of the four miRs. *p* < 0.05 was considered statistically significant.

## 3. Results

### 3.1. Demographic and Clinical Data

Table 1 summarizes the demographic data and clinical characteristics of 99 BC patients. About 80% of the cases were invasive ductal carcinoma (IDC). According to the Union for International Cancer Control (UICC) tumor, node, and metastasis (TNM) Classification and the American Joint Committee on Cancer (AJCC) staging system 2010 [23], almost half of the tumors were in stage III or IV. Distant metastasis was found in 9 patients, while lymph node metastasis was found in 66 patients. In addition, most of the tumors were moderate to high differentiated (G2 = 51.1%; G3 = 30.3%).

The hormone receptor’s status was also evaluated in all samples. Progesterone receptors (PR) were positive in 50.5% of patients, estrogen receptors (ERs) were positive in 48% of patients, and human epidermal growth factor 2 (HER2) overexpression was found in 28.3% of patients (score 3 on immunohistochemistry {IHC} or score 2 on IHC with positive fluorescence in situ hybridization {FISH}). Molecular classification revealed that 40 patients (40.4%) had luminal B BC, 18 (18.2%) had luminal A BC, 13 (13.1%) had HER2-enriched BC, while 18 (18.2%) had triple-negative BC and 10 patients were undetermined.

### 3.2. Correlation of miR Expression with Clinicopathological Data

As shown in Table 1, there were no significant differences in serum miR expression when age, body mass index (BMI), and histological types were considered. Serum miR-155 and miR-10b were significantly upregulated in BC patients with large tumor size (*p* ≤ 0.001) and extensive nodal involvement (*p* ≤ 0.001). Both serum miR-155 and miR-10b (*p* < 0.001) were positively correlated with tumor stage, while the *p*-values of miR-34a and miR-373 in stage I-IV were 0.855 and 0845, respectively, which indicates that miR-34a and miR-373 cannot be used for the diagnosis of BC patients.

Serum miR-155 (*p* < 0.001) and miR-373 (*p* = 0.049) were positively correlated with tumor grade. Serum miR-373 was significantly upregulated in HER2-negative (*p* < 0.001), ER-positive (*p* < 0.005), and PR-positive (*p* < 0.024) BC patients, while serum miR-155 was significantly upregulated in PR-negative (*p* < 0.001) BC patients. We found no significant correlation between serum miRs-155 (*p* = 0.62), miR-10b (*p* = 0.562), miR-373 (*p* = 0.702), and miR-34a (*p* = 0.571) in metastatic (M1) BC patients compared to M0 BC patients. However, we detected a statistical significance when comparing M1 BC patients with the control group.

### 3.3. Serum miR Expression in BC

Circulating serum miR-155 (*p* < 0.001), miR-373 (*p* < 0.001), and miR-10b (*p* < 0.001) were upregulated in the non-metastatic M0 BC group compared to the control group. However, serum miR-34a was downregulated in the BC group compared to the control group (*p* < 0.001) (Figure 2).

### 3.4. Diagnostic Accuracy of the Four miRs

Figure 3 shows the sensitivity versus specificity plots for the four miRs using the ROC curve to detect the diagnostic accuracy of each miR. As shown in Table 2, for serum miR-155, the area under the curve (AUC) was 0.944 (95% confidence interval (CI) = 0.889–0.977; *p* = 0.001). The best cut-off value for BC detection was 7.5, with 86.9% sensitivity and 90% specificity. For serum miR-373, the AUC was 0.948 (95% CI = 0.895–0.979; *p* =0.001). The best cut-off value for BC detection was 10, with 85% sensitivity and 100% specificity. For serum miR-10b, the AUC was 0.768 (95% CI = 0.686–0.838; *p* =0.001). The best cut-off value for BC detection was 9.5, with 60% sensitivity and 93% specificity. For serum miR-34a, the AUC was 0.887 (95% CI = 0.820–0.936; *p* = 0.001). The best cut-off value for BC detection was 10.5, with 91% sensitivity and 75% specificity. ROC curve analysis showed high diagnostic accuracy (AUC = 1.0) when the four miRs were combined (Figure 3).

The area under the curve (AUC): for serum miR-155, the AUC: 0.944 (95% confidence interval (CI) = 0.889–0.977; *p* = 0.001), For serum miR-373, the AUC: 0.948 (95% CI = 0.895–0.979; *p* =0.001), for serum miR-10b, the AUC: 0.768 (95% CI = 0.686–0.838; *p* =0.001), for serum miR-34a, the AUC: 0.887 (95% CI = 0.820–0.936; *p* = 0.001), and (AUC = 1.0) when the four miRs were combined.

## 4. Discussion

Numerous studies have shown that miRs in human plasma and serum are highly resistant to RNase activity [24,25]. miRs might represent an upcoming non-invasive blood-based test for BC screening, diagnosis, and prognosis. In this study, we found that the circulating serum miR-155, miR-373, miR-34a, and miR-10b were significantly dysregulated in the BC group compared to the control groups. This dysregulation was not statistically significant when considering age, BMI, and histologic type, indicating the potential diagnostic role of serum miR-155, miR-373, miR-34a, and miR-10b.

miR-155, an oncogenic miR, is overexpressed in various human malignancies. It plays a critical role in BC by affecting different targets and altering signaling pathways, thereby promoting oncogenic pathways in transformed cells. One of the most important targets in BC tumorigenesis is the Ras homolog family member A (RhoA) protein, which is a member of the Rho guanosine triphosphatase (GTPase) family and is involved in various biological processes, especially epithelial–mesenchymal transition (EMT) and cell migration and invasion. Serum miR-155 might target RhoA, and it is controlled epigenetically by BRCA1 [26,27,28]. In our study, serum miR-155 was upregulated in the BC group compared to the control group. These findings are consistent with some studies [29,30,31,32] and inconsistent with other studies [15,33]. Hamam et al. (2016) explained serum miR downregulation by suggesting that circulating serum miRs are unlikely to be derived from the tumor itself but, instead, reflect generalized homeostatic responses during health and disease [33]. Our study also showed that serum miR-155 is upregulated in BC patients with a large tumor, a high tumor grade, and extensive lymph node metastasis. Serum miR-155 might promote tumor growth by activating tumor-associated macrophages in BC [34]. Our findings are supported by two other separate studies by Chen et al. (2012) and Shaker et al. (2015) The authors reported serum miR-155 upregulation correlated with an advanced tumor stage, a high tumor grade, and lymph node metastasis [30,35].

Regarding tumor biology, our study showed serum miR-155 upregulation in PR-negative tumors which is consistent with other studies [16,32]. However, Zhu et al. (2009) reported higher serum miR-155 expression in PR-positive compared to PR-negative tumors [29]. In addition, we found no significant difference in the HER2 status, which is contradictory to the findings of Nasser et al. (2014) who reported that serum miR-155 is upregulated in HER2/neu-positive patients [16]. Therefore, further studies are required in order to clarify the role of serum miR-155 during BC tumorigenesis and progression. Variations in miR results in tumor samples are caused by technical difficulties to extract miRs from tissue samples, real-time PCR optimization, use of fluctuated genes as an internal reference, the cohort size, and the genetic background of the cohort. Therefore, these variations need to be standardized.

The role of miR-373 in cancer development is contradictory. miR-373 was first identified as a metastasis-promoting miR in BC. In a study by Yan et al. (2011), >30 proteins were involved in serum miR-373–regulated cancer cell invasion and metastasis [36]. However, miR-373 plays a role in inhibiting cancer cell migration and invasion through E-cadherin upregulation [37]. Our study detected serum miR-373 upregulation in the BC group compared to the control group. In addition, circulating serum miR-373 was upregulated in HER2-negative, ER-positive, and PR-positive BC tumors. Our findings are consistent with other studies [15,31,38]. Eichelser et al. (2013) reported high serum miR-373 upregulation in M0 BC patients compared to healthy women but found no significance in M1 BC patients [15]. The authors also reported that high serum miR-373 upregulation is correlated with a negative HER2 status of the primary tumor.

Evidence showed that by being a direct p53 target, miR-34a acts as a tumor-suppressor gene, and its upregulation induces apoptosis and cell cycle arrest [39,40,41]. Loss of 1p36 and CpG methylation of the miR-34a promoter leads to BC tumorigenesis by inhibiting cancer cell invasion, both in vitro and in vivo, and inhibiting fos-related antigen-1 in BC cells [42,43]. Our findings showed serum miR-34a downregulation in the BC group compared to the control group without any correlation with clinicopathological data. These findings are consistent with other studies [44,45]. However, Eichelser et al. (2013) reported that serum miR-34a is upregulated and correlated with triple-negative and HER2-positive BC patients [15]. Interestingly, although serum miR-34a was downregulated in our study, the miR-34a level was low in M1 BC patients compared to M0 BC patients.

miR-10b is significantly upregulated in patients with metastasis and cancer cell invasion and is a promising candidate for the development of new antimetastasis agents [46]. Twist, a key transcription factor that induces EMT and favors metastases, upregulates miR-10b in vitro [47]. Chen et al. (2017) reported that the potential functional single-nucleotide polymorphism (SNP) (rs4078756) in the miR-10b promoter region might contribute to BC among Chinese women [48]. Our study reported serum miR-10b upregulation in the BC group compared to the control group, which is consistent with previous studies [38,49]. Although serum miR-10b was upregulated in M1 BC patients, the increase was not significant compared to M0 BC patients. Studies have reported serum miR-10b downregulation or no change in BC patients compared to healthy women [50,51].

In our study, while the four miRs were dysregulated in BC patients, their levels did not differ significantly between M1 and M0 groups. Shaker et al. (2015) reported high serum miR-155 upregulation in M1 compared to M0 BC patients [30]. However, Hagrass et al. (2015) found different miR-155, miR-34a, and miR-10b levels in M1 compared to M0 BC patients [45]. In addition, Eichelser et al. (2013) did not find any significant difference in serum miR-34a and miR-373 levels but found miRNA-155 upregulation in M0 compared to M1 BC patients [15]. All these findings suggest that these miRs might be useful in early BC detection, but their role in predicting metastasis still needs further investigation.

## 5. Conclusions

Four miRNAs (miR-155, miR-373, miR-10b, and miR-34a) have a diagnostic ability to distinguish BC patients from healthy women at the early stage of the disease. Serum miR-155 and miR-373 are positively correlated with tumor grade. Serum miR-373 is upregulated in HER2-negative, ER-negative, and PR-positive BC patients. Multi-institutional studies are required in order to evaluate the use of miRs as reliable biomarkers for early BC detection in women from different cohorts.

## Figures and Tables

**Figure 1 diagnostics-12-00789-f001:**
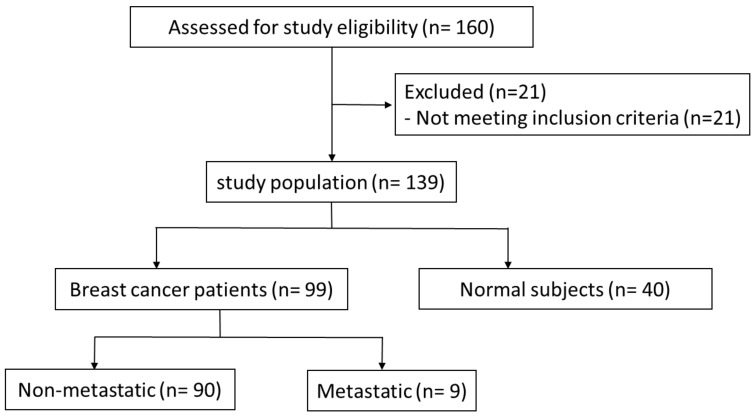
Stats of Primary BC patients and healthy volunteers.

**Figure 2 diagnostics-12-00789-f002:**
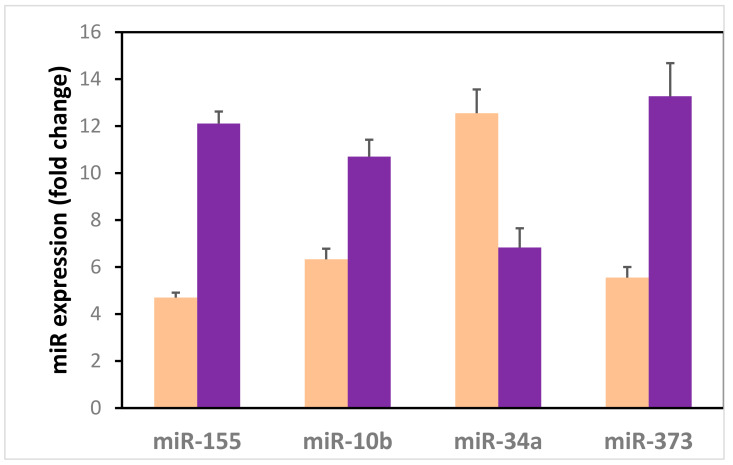
Circulating serum microRNAs in the non-metastatic BC group in contrast to the control group. 
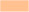
 Control group 
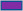
 BC group.

**Figure 3 diagnostics-12-00789-f003:**
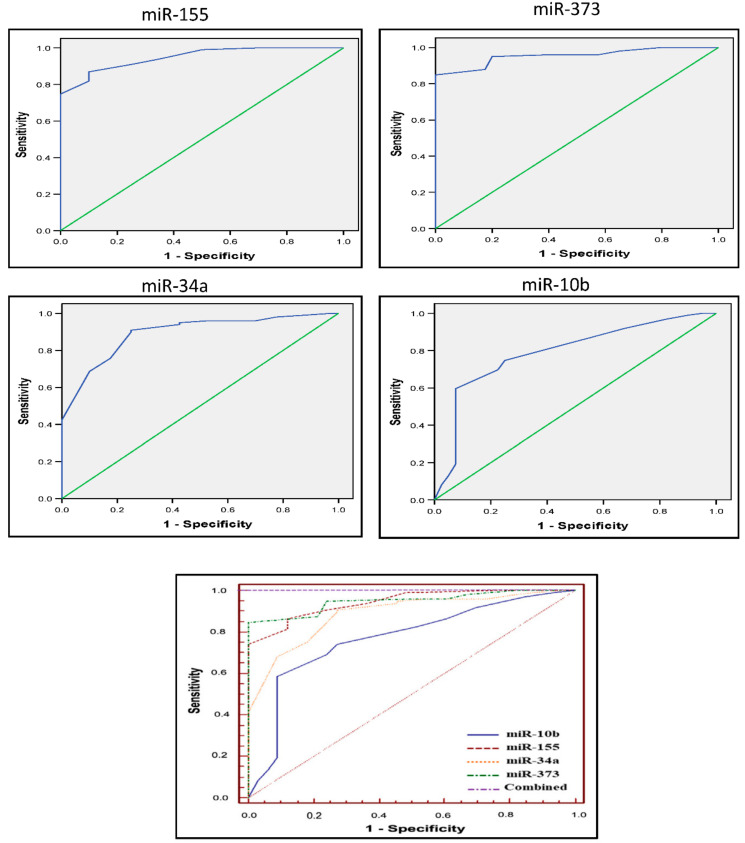
Sensitivity versus specificity plots for the four miRs using the ROC curve.

**Table 1 diagnostics-12-00789-t001:** Correlation between FCs of miR-155, miR-10b, miR-34a, and miR-373 and clinicopathological data of BC patients.

Patient Presentation		Relative Expression of miRs and *p*-Value								
		*N*	miR-155	*p*	miR-10b	*p*	miR-34a	*p*	miR-373	*p*
			Mean (SD)		Mean (SD)		Mean (SD)		Mean (SD)	
Age	<40	9	11.67 (4.15)	0.699	12.33 (3.32)	0.204	5.78 (3.03)	0.275	13.44 (3.58)	0.872
	≥40	90	12.16 (3.55)		10.53 (4.09)		6.93 (3.01)		13.26 (3.32)	
BMI	Obese	35	11.77 (3.34)	0.489	11.43 (3.99)	0.184	7.11 (3.21)	0.488	12.6 (3.33)	0.137
	Normal/ overweight	64	12.3 (3.73)		10.3 (4.05)		6.67 (2.92)		13.64 (3.29)	
Grade	1	18	6.89 (1.61)	<0.001	9.94 (4.4)	0.43	6.44 (2.38)	0.839	12.61 (3.58)	0.049
	2	51	12.53 (2.64)		10.53 (4.35)		6.92 (2.9)		14.06 (2.89)	
	3	30	14.53 (2.58)		11.43 (3.2)		6.9 (3.58)		12.33 (3.62)	
LN stage	0	33	10.36 (3.55)	0.001	6.39 (2.14)	<0.001	6.18 (2.11)	0.516	13.79 (3.57)	0.096
	1	27	11.62 (2.81)		11 (2.68)		7.19 (3.81)		13.33 (2.65)	
	2	33	14.00 (2.90)		13.79 (2.26)		7.09 (2.84)		12.3 (3.6)	
	3	6	16.80 (1.07)		16 (1.27)		7.33 (4.23)		15.5 (1.23)	
Tumor size	1	25	8.04 (2.89)	<0.001	9.24 (3.99)	0.001	6.96 (2.48)	0.968	12.84 (3.85)	0.749
	2	58	13.17 (2.69)		10.45 (4.01)		6.78 (3.21)		13.45 (3.2)	
	3	16	14.63 (2.13)		13.88 (2.39)		6.81 (3.23)		13.31 (3.01)	
Metastasis	0	90	12.04 (3.64)	0.562	10.62 (4.03)	0.563	6.73 (2.88)	0.324	13.33 (3.28)	0.569
	1	9	12.78 (3.15)		11.44 (4.3)		7.78 (4.24)		12.67 (3.91)	
Stage	I	14	7.57 (3.28)	<0.001	6.86 (2.25)	<0.001	6.64 (1.87)	0.855	13.14 (4.11)	0.845
	II	36	12.97 (3.07)		8.35 (3.37)		6.54 (3.14)		13.65 (2.9)	
	III	40	12.63 (3.2)		13.88 (2.29)		7.1 (3.08)		13.08 (3.32)	
	IV	9	6.89 (1.61)		12.38 (3.5)		7.13 (4.02)		12.75 (4.17)	
Her2/neu receptor	Positive	27	11.59 (4.02)	0.382	10.63 (4.25)	0.96	6.41 (2.06)	0.365	10.78 (4.33)	<0.001
	Negative	71	12.31 (3.45)		10.68 (4)		7.03 (3.3)		14.23 (2.28)	
ER status	Positive	48	12.25 (3.59)	0.711	10.63 (4.04)	0.865	6.6 (2.85)	0.476	14.23 (3.1)	0.005
	Negative	51	11.98 (3.62)		10.76 (4.09)		7.04 (3.18)		12.37 (3.3)	
PR status	Negative	41	15.23 (2.31)	0.001	10.78 (4.11)	0.315	6.76 (3.42)	0.863	12.78 (3.24)	0.024
	Positive	50	10.54 (2.83)		10.96 (4)		6.96 (2.87)		14.04 (3.14)	
Pathology type	IDC	80	12.08 (3.48)	0.72	10.68 (4.06)	0.96	6.69 (3.14)	0.455	13.16 (3.35)	0.066
	ILC	11	12.82 (4.12)		11 (4.03)		7.91 (2.39)		15.18 (1.72)	
	Mixed	8	11.5 (4.28)		10.5 (4.44)		6.75 (2.44)		11.75 (3.88)	
Control vs. M1 cases	Control	40	4.7 (2.38)	<0.001	6.33 (3.63)	0.003	12.55 (3.37)	0.004	5.55 (2.51)	<0.001
	M1 cases	9	12.78 (3.15)		11.44 (4.3)		7.78 (4.24)		12.67 (3.91	

FCs, fold changes; miR, microRNA; BC, breast cancer; SD, standard deviation; BMI, body mass index; HER2, human epidermal growth factor 2; ER, estrogen receptor; PR, progesterone receptor; IDC, invasive ductal carcinoma; ILC, invasive lobular carcinoma; LN, lymph nodes; M1, metastasis; *p* value > 0.05 is not significant; *p* value < 0.05 is significant; *p* value < 0.001 is highly significant.

**Table 2 diagnostics-12-00789-t002:** AUC for the four miRs included in the study.

Test Result Variable(s)	AUC	Cut-Off Value	Sensitivity	Specificity	Std. Error ^a^	Asymptotic Significance ^b^	Asymptotic 95% CI	
							Lower	Upper
miR-155	0.944	7.5	86.9	90	0.019	0.001	0.889	0.977
miR-373	0.948	10	85	100	0.018	0.001	0.895	0.979
miR-10b	0.768	9.5	60	93	0.043	0.001	0.686	0.838
miR-34a	0.887	10.5	91	75	0.039	0.001	0.820	0.936

AUC, the area under the curve; miR, microRNA; CI, Confidence Interval. The test result variable(s) (miR155, miR10b, miR34a, and miR373) had at least one tie between the positive and negative actual state groups. ^a^ Nonparametric assumption. ^b^ Null hypothesis (true area = 0.5).

## Data Availability

Not applicable.

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
