# Peer review of "Evaluation of Expressed MicroRNAs as Prospective Biomarkers for Detection of Breast Cancer"

_diagnostics, 2022, doi:10.3390/diagnostics12040789_

Round 1
Reviewer 1 Report
In this manuscript, the authors evaluated the expression levels of circulating serum miRNAs (miR-155, 373, 10b, and 34a) for early detection of breast cancer (BC) by using quantitative real-time polymerase chain reaction. In addition, the sensitivity and specificity were estimated by performing the statistical analyses. Overall, the work was completely executed by analyzing a number of BC patients and healthy volunteers (n=160). However, there are many issues that should be resolved before publication, and thus I suggest the acceptance of this manuscript only after accommodating the following comments.
- In the abstract, the susceptibility should be changed into sensitivity. In addition, there are many typos, and the authors need to get their manuscript edited by the native speakers.
- The authors utilized serum miRNAs, but exosome miRNAs have emerged as the promising biomarkers for BC diagnosis. Thus, the authors need to discuss exosome miRNAs for the diagnosis of BC.
- There are many miRNAs for the diagnosis of BC, but four serum miRs (miR-155, miR-373, miR-10, and miR-34a) were selected. The authors need to provide the criteria for the selection of four serum miRNAs.
- In Table 2 and 3, the authors need to provide the results obtained when four miRNAs were combined. In addition, the authors need to try with different combinations of miRNAs.
- In Figure 3, the last figure was differently drawn, and thus the authors need to draw them in the consistent manner.
- To demonstrate that the early detection of BC is feasible, the authors need to show the results according to the different cancer staging (Stage 1-4).
- There are many papers dealing with serum miRNAs for BC diagnosis, and the novelty and key results of this work are unclear. Therefore, the authors need to clearly explain what the difference and main results are in this work.
Author Response
Dear Editor and Reviewers,
Manuscript ID: diagnostics-1622330
Type of manuscript: Article
Title: Evaluation of expressed microRNAs as prospective biomarkers for detection of breast cancer
Thank you very much for your fruitful comments and recommendations, the requested comments are already included and highlighted in the main manuscript, here, the responses to each reviewer comments separately.
Reply to reviewer 1 Comments
In this manuscript, the authors evaluated the expression levels of circulating serum miRNAs (miR-155, 373, 10b, and 34a) for early detection of breast cancer (BC) by using a quantitative real-time polymerase chain reaction. In addition, the sensitivity and specificity were estimated by performing the statistical analyses. Overall, the work was completely executed by analyzing a number of BC patients and healthy volunteers (n=160). However, there are many issues that should be resolved before publication, and thus I suggest the acceptance of this manuscript only after accommodating the following comments.
Point 1: In the abstract, the susceptibility should be changed into sensitivity. In addition, there are many typos, and the authors need to get their manuscript edited by native speakers.
Response: The susceptibility has been changed into sensitivity in the abstract and highlighted in the manuscript.
Point 2: The authors utilized serum miRNAs, but exosome miRNAs have emerged as promising biomarkers for BC diagnosis. Thus, the authors need to discuss exosome miRNAs for the diagnosis of BC.
Response: The definition of exosome was added in the manuscript in the introduction see the attached manuscript.
Point 3: 55, miR-373, miR-10, and miR-34a) were selected. The authors need to provide the criteria for the selection of four serum miRNAs.
Response: The criteria for the choice of miRNAs were added in the manuscript in the introduction see the attached manuscript.
Point 4: In Tables 2 and 3, the authors need to provide the results obtained when four miRNAs were combined. In addition, the authors need to try with different combinations of miRNAs.
Response: Results of combinations of miRNAs were provided in figure 3
Point 5: In Figure 3, the last figure was differently drawn, and thus the authors need to draw them in a consistent manner.
Response: The last figure was deleted from Figure 3 due to inconsistency and heterogeneity with other figures.
Point 6: There are many papers dealing with serum miRNAs for BC diagnosis, and the novelty and key results of this work are unclear. Therefore, the authors need to clearly explain what the difference and main results are in this work.
Response: The novelty is represented by the critical role of circulating serum miR-155, miR-373, miR-10b, and miR-34a as potential biomarkers for early breast cancer detection in Egyptian patients, and observing their combination, which showed high sensitivity and specificity to improve the diagnosis.
Thank you very much for your comment, all comments have been covered in the manuscript.
Reviewer 2 Report
1. The article is devoted to the determination of miR-155, miR-373, miR-10b, and miR-34a for the early diagnosis of breast cancer. However, the study included patients up to stage 4, which cannot be accurately attributed to early diagnosis. You need to make changes to the title or change the concept. 2. Didn't see miR content for healthy control. It should be added to the table as a separate line. 3. Compared to what are the statistical differences in Table 1 indicated? 4. In figure 2, you need to label the columns. 5. MiR-155 misprint in table 2. 6. Did the authors take into account simultaneously 4 miRs or individual combinations of miRs regarding diagnostic significance? Table 2 does not show the corresponding area under the curve for simultaneous accounting for 4 miR.
Author Response
Dear Editor and Reviewers,
Manuscript ID: diagnostics-1622330
Type of manuscript: Article
Title: Evaluation of expressed microRNAs as prospective biomarkers for detection of breast cancer
Thank you very much for your fruitful comments and recommendations, the requested comments are already included and highlighted in the main manuscript, here, the responses to each reviewer comments separately.
Reply to reviewer 2 Comments
Point 1: The article is devoted to the determination of miR-155, miR-373, miR-10b, and miR-34a for the early diagnosis of breast cancer. However, the study included patients up to stage 4, which cannot be accurately attributed to early diagnosis. You need to make changes to the title or change the concept.
Response: The manuscript title was changed to be as follows; Evaluation of expressed microRNAs as prospective biomarkers for detection of breast cancer
Point 2: Did not see miR content for healthy control. It should be added to the table as a separate line.
Response: miR content for healthy control were measured and listed in Table 3
Point 3: Compared to what are the statistical differences in Table 1 indicated?
Response: This is a Correlation between micro and clinicopathological data of breast cancer patients
Point 4: In figure 2, you need to label the columns.
Response: Columns were labeled
Point 5: MiR-155 misprint in table 2.
Response: MiR-155 misprint was corrected
Point 6: Did the authors take into account simultaneously 4 miRs or individual combinations of miRs regarding diagnostic significance? Table 2 does not show the corresponding area under the curve for simultaneous accounting for 4 miR.
Response: A combination of circulating serum miR-155, miR-373, miR-10b, and miR-34a as potential biomarkers for early BC detection in Egyptian patients, showed high sensitivity and specificity. In Egypt, there were many risk factors for breast cancer and in the last years, there was a tangible improvement of both screening and surveillance strategies of a breast cancer diagnosis.
Thank you very much for your comment, all comments have been covered in the manuscript.
Round 2
Reviewer 1 Report
The authors responded to the issues raised by the reviewers. However, some of the issues were not still resolved. Thus, I suggest the acceptance of this manuscript only after accommodating the following comments.
- The authors need to get their manuscript edited by the native speakers. There are many awkward sentences. One example is as follows: “increased serum level of exosomal miRNA-373 in BC patients compared with healthy volunteers and suggested an association of this miRNA with aggressive BC. In addition, there are many typos: one example is as follows : “ROC curve analysis showed high diagnostic accuracy when the four miRNAs were combined (Fig. 3 and Table 2)”. In Figure 3 and Table 2, there are no information of AUC when the four miRNAs were combined.
- The authors need to add more detailed description about Table 1. It is unclear what the meaning of P-value in Table 1. For example, the p-values of miR-155, miR-10b, miR-34a, and miR-373 in stage I-IV were < 0.001, <0.001, 0.855, and 0845, respectively, which indicates that miR-34a and miR-373 cannot be used for the diagnosis of BC patients. In addition, no data were provided when four miRNAs were combined.
- In Table 1, the full names of N, T, and M should be provided.
- The authors need to include AUC when the four miRNAs were combined. In addition, there are no results of different combinations of miRNAs in Figure 3. Thus, the authors need to include these results in Table 2 and 3.
- It would be better to merge Table 2 and 3 into one table.
Author Response
Dear Editor and Reviewers,
Manuscript ID: diagnostics-1622330
Type of manuscript: Article
Title: Evaluation of expressed microRNAs as prospective biomarkers for detection of breast cancer
Thank you very much for your fruitful comments and recommendations, the requested comments are already included and highlighted in the main manuscript, here, the responses to each reviewer comments separately.
Reply to reviewers Comments
Reviewer 1:
Comments and Suggestions for Authors
The authors responded to the issues raised by the reviewers. However, some of the issues were not still resolved. Thus, I suggest the acceptance of this manuscript only after accommodating the following comments.
Point 1
The authors need to get their manuscript edited by native speakers. There are many awkward sentences. One example is as follows: “increased serum level of exosomal miRNA-373 in BC patients compared with healthy volunteers and suggested an association of this miRNA with aggressive BC.
Response:
Thank you for your comment. The manuscript has been edited and revised.
Point 2
In addition, there are many typos: one example is as follows: “ROC curve analysis showed high diagnostic accuracy when the four miRNAs were combined
(Fig. 3 and Table 2)”. In Figure 3 and Table 2, there is no information of AUC when the four miRNAs were combined.
Response:
Thank you for your critical comment. We added the figure showing of AUC when the four miRNAs were combined, and the sentence was corrected to “ROC curve analysis showed high diagnostic accuracy when the four miRNAs were combined (Fig. 3)”
Point 3
The authors need to add a more detailed description about Table 1. It is unclear what the meaning of P-value in Table 1. For example, the p-values of miR-155, miR-10b, miR-34a, and miR-373 in stage I-IV were < 0.001, <0.001, 0.855, and 0845, respectively, which indicates that miR-34a and miR-373 cannot be used for the diagnosis of BC patients.
Response:
Thank you for your comment. We added the following paragraph to (Both serum miR-155 and miR-10b (P < 0.001) were positively correlated with tumor stage, while the p-values of miR-34a and miR-373 in stage I-IV were 0.855, and 0845, respectively, which indicates that miR-34a and miR-373 cannot be used for the diagnosis of BC patients.) under the title 3.2. Correlation of miR expression with clinicopathological data
Point 4
In Table 1, the full names of N, T, and M should be provided.
Response:
Thank you for your comment. We added the full names of N, T, and M in Table 1.
Point 5
The authors need to include AUC when the four miRNAs were combined.
Response:
Thank you for your comment. We have added a figure to figure 3 showing AUC when the four miRNAs were combined.
Point 6
In addition, there are no results of different combinations of miRNAs in Figure 3. Thus, the authors need to include these results in Tables 2 and 3.
Response:
We kindly asked the Editor to give us an extra two days after the given deadline, but unfortunately, we did not receive any reply Now we are waiting for the editor’s reply while we are preparing the statistical analysis for the combinations of 2 miRNAs, or 3. But we had to send the revised manuscript for respecting the given deadline
Point 7
It would be better to merge Tables 2 and 3 into one table.
Response:
Thank you for your comment. We could not do that as Tables 1 and 3 were merged into one table as suggested by reviewer 2.
Thank you very much

Reviewer 2 Report
1) In Table 1, add the content of miRNA as the top line in the norm, so it is more convenient to perceive information than to compare with Table 3. 2) I did not receive an answer to the last question. So, if all 4 miRNAs change in breast cancer, then it is necessary to establish threshold values relative to which the content of individual miRNAs increases or decreases. If we simultaneously take into account the increase in miR-155, miR-10b, and miR-373 and the decrease in miR-34a, how will the sensitivity and specificity indicators change? If you combine 2 miRNAs, or 3?
Author Response
Dear Editor and Reviewers,
Manuscript ID: diagnostics-1622330
Type of manuscript: Article
Title: Evaluation of expressed microRNAs as prospective biomarkers for detection of breast cancer
Thank you very much for your fruitful comments and recommendations, the requested comments are already included and highlighted in the main manuscript, here, the responses to each reviewer comments separately.
Reply to reviewers Comments
Reviewer 2:
Comments and Suggestions for Authors
Point 1
In Table 1, add the content of miRNA as the top line in the norm, so it is more convenient to perceive information than to compare with Table 3.
Response:
Thank you for your comment. Table 1 and 3 were merged into one table
Point 2
2) I did not receive an answer to the last question. So, if all 4 miRNAs change in breast cancer, then it is necessary to establish threshold values relative to which the content of individual miRNAs increases or decreases. If we simultaneously take into account the increase in miR-155, miR-10b, and miR-373 and the decrease in miR-34a, how will the sensitivity and specificity indicators change? If you combine 2 miRNAs, or 3?
Response:
We kindly asked the Editor to give us an extra two days after the given deadline, but unfortunately, we did not receive any reply Now we are waiting for the editor’s reply while we are preparing the statistical analysis for the combinations of 2 miRNAs, or 3.
But we had to send the revised manuscript for respecting the given deadline
Thank you very much

This manuscript is a resubmission of an earlier submission. The following is a list of the peer review reports and author responses from that submission.